# Experimental Study on the Properties of Mixed-Fiber Concrete Shield Tunnel Segments Subjected to High Temperatures

**Yajun Zhang** [1,2]**, Yao Wang** [1,*] **and Zhaoqing Ren** [2]

1   College of Architecture and Engineering, Yancheng Polytechnic College, Yancheng 224005, China
2   Jiangsu Key Laboratory of Environmental Impact and Structural Safety in Engineering,
    China University of Mining and Technology, Xuzhou 221008, China
*   Correspondence: yaowang@yctei.edu.cn

**Abstract:** In order to study the mechanical and damage behavior of concrete shield tunnel segments under a high temperature, two self-compacting concrete and three mixed-fiber (steel and polypropylene fiber) self-compacting concrete test blocks were designed. The influence of several key factors, including fire duration, pre-loading, and concrete type, on the fire behavior of concrete shield tunnel segments were studied. The results show that the type of fiber and pre-loading have an important influence on crack development in concrete shield tunnel segments. Compared with undoped segments, cracks in segments with steel fibers and polypropylene fibers appeared later, and the average crack spacing decreased. The pre-loading has an important effect on the vertical deformation before and after the temperature rise. As the load level increases, so does the deformation after the temperature rise. The influence of the initial load level should be considered when designing the fire resistance of the segment.

**Keywords:** concrete shield tunnel segments; fire tests; mixed-fiber concrete; pre-loading





## 1. Introduction

As one of the main disasters faced by tunnels, fires often occur in various types of tunnels around the world. The damage of the lining structure under fire not only seriously reduces the bearing capacity of the lining structure, but even causes the tunnel to collapse due to the deterioration of the mechanical properties of the lining concrete and the reduction in the thickness of the lining section caused by bursting. Therefore, it is of great theoretical value and practical significance to study the failure mode of tunnel lining segments under a high fire temperature.

For shield tunnel segments, the fire test research has achieved very rich results. For small-scale tests, the literature [1–3] analyzes the temperature field distribution, relative humidity, structural deformation, and bearing capacity of concrete lining structures under fire. For full-scale structure and field tests, scholars have also conducted research. In 2011, Zhou [4] obtained the central vertical temperature distribution at different sections of the tunnel under different working conditions. In 2014, Lai et al. [5] conducted large-scale model tests to study the temperature field distribution and failure of highway tunnel lining structures under different moisture reductions. The results show that, although moisture can reduce the temperature inside the lining, it aggravates the damage of the lining components. In 2018, Pan et al. [6] studied the mechanical properties and damage laws of shield tunnel lining structures under the action of high temperature fire from two different fire scenarios, uniform and non-uniform, analyzed the physical damage forms (bursting, cracking, and dehydration) and the development laws of tunnel lining concrete under the action of fire, and explained the mechanism of high-temperature bursting of lining concrete.

During the experiment, the researchers found that ordinary concrete has poor mechanical properties at high temperatures and is very prone to bursting behavior. Therefore,

it is improved by adding fibers to the concrete. Studies have shown that the addition of fibers can reduce the explosion and spalling of cementitious materials and improve their mechanical properties after high temperatures [7–9]. Scholars applied fiber concrete to lining structures; the results show that the incorporation of fibers has a significant inhibitory effect on the high-temperature bursting of lining concrete [10–16]. In order to study the fire behavior of concrete lining structures more realistically, scholars studied the thermodynamic coupling behavior of shield tunnel lining structures under high fire temperatures [12,17], and the results show that the pre-loading significantly affected the high temperature failure mode and mechanical properties of the lining segment.

It can be seen from the above that many studies have been carried out on the high temperature mechanical behavior of shield tunnel lining structural systems, but there are still few research results on the mechanical properties of mixed-fiber thermodynamic coupling behavior in fire conditions. In order to deeply grasp the fire characteristics and safety of the prefabricated shield tunnel lining structure system, it is necessary to analyze and study the fire behavior of mixed-fiber concrete segment heating.

In this paper, the fire tests of five concrete shield tunnel segments were carried out, including two self-compacting concrete segments (RC—1,2) and three self-compacting concrete segments with mixed fibers (steel fiber and polypropylene fiber, HFRC—1,2,3). The temperature field distribution, bursting, deformation, and crack development trends of the segments were studied during different loading and heating processes.

## 2. Test

### 2.1. Test Specimens

In this study, five concrete shield tunnel segments with a thickness of 17.5 cm, width of 60 cm, arch of 67.5°, and outer arch radius of 155 cm were used. The thickness of the concrete protective layer was 25 mm. The concrete mix ratio is shown in Table 1, and 30 kg/m$^3$ steel fibers and 2 kg/m$^3$ polypropylene fibers were added to the mixed-fiber concrete. After curing, the compressive strength of the concrete and mixed-fiber concrete was 52.3 MPa and 50.6 MPa, respectively. The ring main bar adopted was HRB400, the diameter of the steel bar was 10 mm, the yield strength was 424.6 MPa, and the ultimate strength was 605.9 MPa. The stirrup adopted was HRB400, the diameter was 6 mm, the spacing was 100 mm, the yield strength was 417.0 MPa, and the ultimate strength is 565.0 MPa, as shown in Figure 1. The specimens were cured in the open air at room temperature for more than 600 days.

**Table 1.** Mixing proportion of the concrete (kg/m$^3$).

| Water | Cement | Sand | Aggregate | Fly Ash | Admixtures |
|-------|--------|------|-----------|---------|------------|
| 180 | 400 | 765 | 832 | 160 | 4.5 |

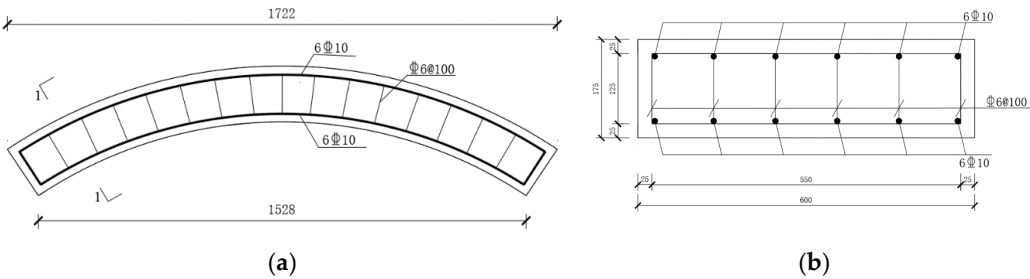

(a)                              (b)

**Figure 1.** Size and reinforcement of test segment (Unit: mm): (**a**) Segment size and reinforcement method; (**b**) 1–1 profile view.

### 2.2. Loading Methods

As shown in Figure 2, the reaction frame of this test was composed of a jack, horizontal reaction frame, a vertical reaction frame, a distribution beam, and a support (hinge support). A high-temperature test furnace was used under the segment.

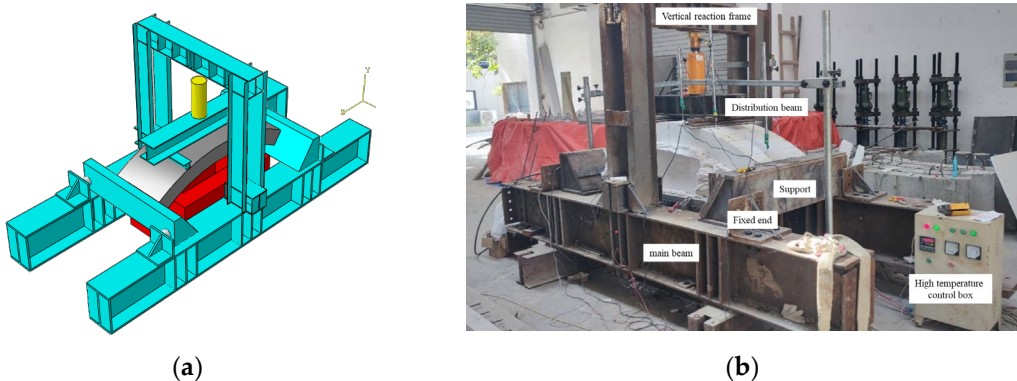

| (a) | (b) |

**Figure 2.** Test set-up diagram at high temperature: (**a**) Three-dimensional drawing of the test set-up; (**b**) Field drawings of the test set-up.

The loading method was mainly in accordance with the GB/T 22082-2017 [18]: the vertical load of the lining segment was loaded centrally at two points, and the loading point was located at the third equal point of the segment specimen, as shown in Figure 3.

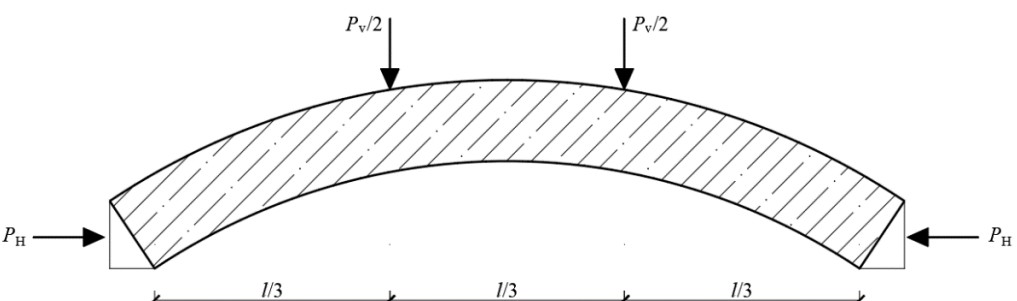

**Figure 3.** Loading method.

The test load was taken according to the theoretical ultimate bearing capacity of 0.15~0.5 of the test tunnel segment at room temperature, so that the segment was in a moderate working state in the test, taking 60 kN, 120 kN, and 180 kN. The grouping of specimens is detailed in Table 2. During the heating process, the pre-loaded load was not allowed to drop. When the applied load was reduced by 10 kN, it was restored to its original value using the jack. During the cooling process, the preload changes naturally.

**Table 2.** The grouping information and furnace temperature of each test segment.

| ID | Pre-Loading(kN) (Pv) | Fire Duration | Maximum Temperature (°C) | Temperature at Ceasefire (°C) | Temperature at the End (°C) |
|---|---|---|---|---|---|
| RC—1 | 60 | 240 | 760 | 754 | 183 |
| RC—2 | 120 | 180 | 902 | 898 | 184 |
| HFRC—3 | 60 | 180 | 803 | 803 | 183 |
| HFRC—4 | 120 | 180 | 813 | 813 | 171 |
| HFRC—5 | 180 | 150 | 798 | 798 | 130 |

Note: RC means self-compacting concrete segments; HFRC means self-compacting concrete segments with mixed fibers.

### 2.3. Temperature, Displacement Measurement, and Horizontal Support Reaction Force

Two furnace thermocouple measurement points were arranged in the high-temperature furnace cavity at the bottom of the test segment. As shown in Figure 4a, three measurement points were arranged along the test segment 1/4 and 3/4, which are numbered TA–TC. The concrete thermocouple arrangement at each measurement point pre-embedded six thermocouple lines along the thickness of the segment, of which the first thermocouple line was placed 10 mm away from the fire surface, the sixth thermocouple line was placed 15 mm from the unfired surface, and the remaining measurement points were arranged every 30 mm, which is indicated by the numbers C—1~C—6. The steel bar thermocouple arrangement tied the thermocouple wire to the steel bar. Numbers SC—1 and SC—2 are the segment top steel bar temperature, while numbers SC—3 and SC—4 are the bottom steel bar temperature of the segment, which is shown in Figure 4b.

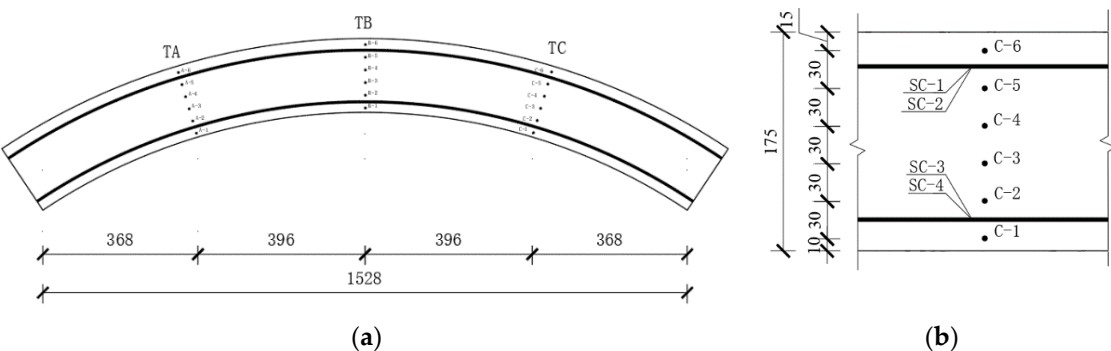

(**a**)                                                                                            (**b**)

**Figure 4.** Layout plan of segmented thermocouple (Unit: mm): (**a**) Plane figure; (**b**) Measurement point arrangement in the cross-section.

As shown in Figure 5a, the six vertical displacement measurement points of the segment were arranged, and the measurement points were symmetrically arranged with two measurement points, V3 and V4, in the middle of the span, the measurement points V2 and V5 were arranged at the cross-sectional position of the third equinox, and two vertical displacement measurement points, V1 and V6, were arranged near the support.

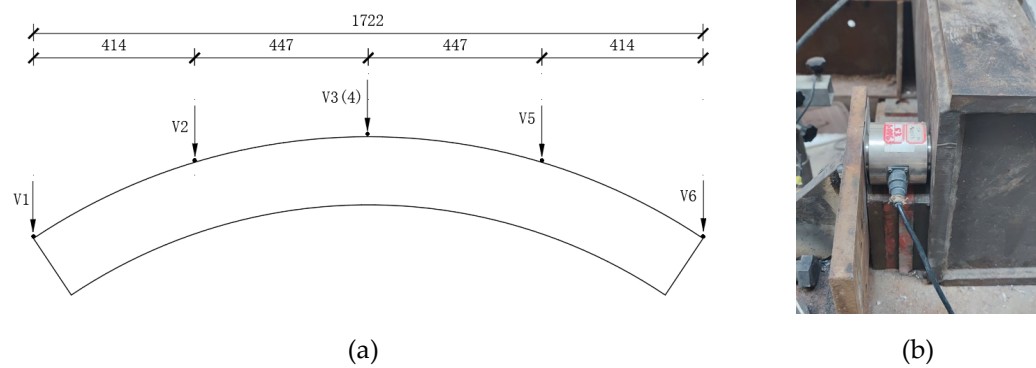

(a)                                                                                            (b)

**Figure 5.** (**a**) Layout of displacement measurement points; (**b**) Horizontal support reaction force measuring device.

As shown in Figure 5b, the force transducer was placed at the support to measure the horizontal support reaction force.

### 2.4. Fire Conditions

In this paper, a furnace was specially designed and built to heat the test specimens (Figure 2). The heating curve refers to ISO834, and when the specified time was reached, the furnace was turned off and cooled naturally. It is worth pointing out that ISO834 was

chosen as the reference fire curve considering the actual working capacity of the furnace, the fire conditions, and the comparison with other test results, but the ISO834 curve does not represent the fire temperature of the underground tunnel due to the rapid rise in the fire temperature in the enclosed space. The furnace temperature results of the five test segments are shown in Table 2 and Figure 6.

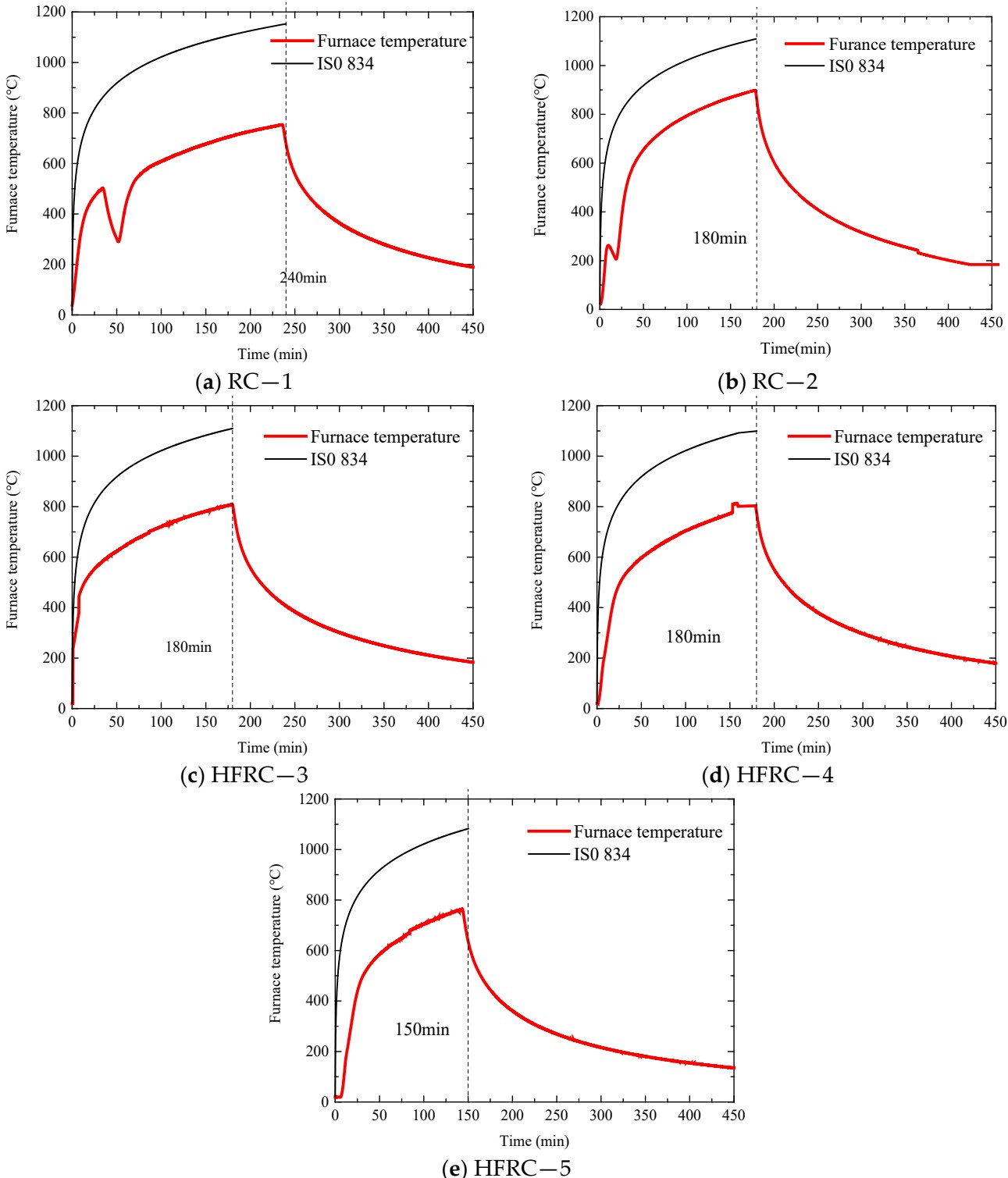

**Figure 6.** Furnace temperature–time curves: (**a**) RC−1; (**b**) RC−2; (**c**) HFRC−3; (**d**) HFRC−4; and (**e**) HFRC−5.

## 3. Results

### 3.1. Cracking Patterns

(1)　Segment RC−1

After 60 min, water vapor began to overflow at the fitting point of the segment and the sealing plate, and multiple cracks appeared on both sides of the segment. After 65 min, the crack developed vertically to the vault, and a penetrating crack appeared, and at the same time, water seepage began to appear on the backfire side of the segment. After 127 min, the water seepage phenomenon on the side of the segment gradually decreased. At 130 min, cracks appeared at the arch foot of the segment. It is worth pointing out that due to the slow heating rate of the test, RC—1 did not undergo significant bursting. The crack diagram is shown in Figure 7.

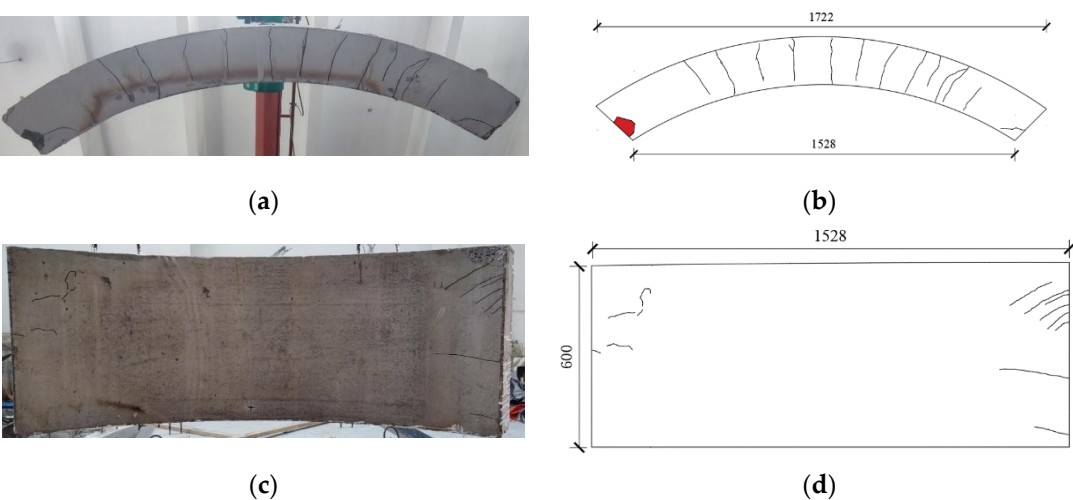

(**a**)　　　　　　　　　　　　　　　　　　　　　　(**b**)

(**c**)　　　　　　　　　　　　　　　　　　　　　　(**d**)

**Figure 7.** Failure modes of Segment RC—1: (**a**) Cracking pattern; (**b**) Crack pattern on the side surface; (**c**) Bottom surface; and (**d**) Cracking pattern on the bottom surface.

Due to the vertical load and temperature stress, the joint part of the vault produces large negative bending due to the extrusion of the adjacent segment, so that it is crushed and damaged, and the crack develops to the middle of the span, accompanied by a large joint opening angle. It can be concluded that, due to the decrease in concrete and steel properties at high temperatures, the concrete at the foot of the arch (the joint part) is easy to crush and fails for the segment of the vault in the shield tunnel system. Therefore, for the shield tunnel structure in the existing project, the fire protection and stiffness at the segments on both sides of the vault should be strengthened to reduce the risk of crushing the concrete of the joint bolt hole.

(2)　Segment RC—2

As shown in Figure 8, cracks began to appear at the loading point when the segment was loaded to 90 kN, and cracks appeared in the middle of the span when loaded to 120 kN, and, after the load is stabilized, it began to heat up. After 20 min of heating, a large number of cracks began to appear in the loading point and span, and the cracks caused by the load developed to the top. At 40 min, water stains appeared at the top of the support and segment, and penetrating cracks appeared at the loading point. At 60 min, blisters seeped out of the crack and water vapor appeared. After 120 min, cracks began to appear at the support, water vapor gradually decreased, and cracks appeared in the vault.

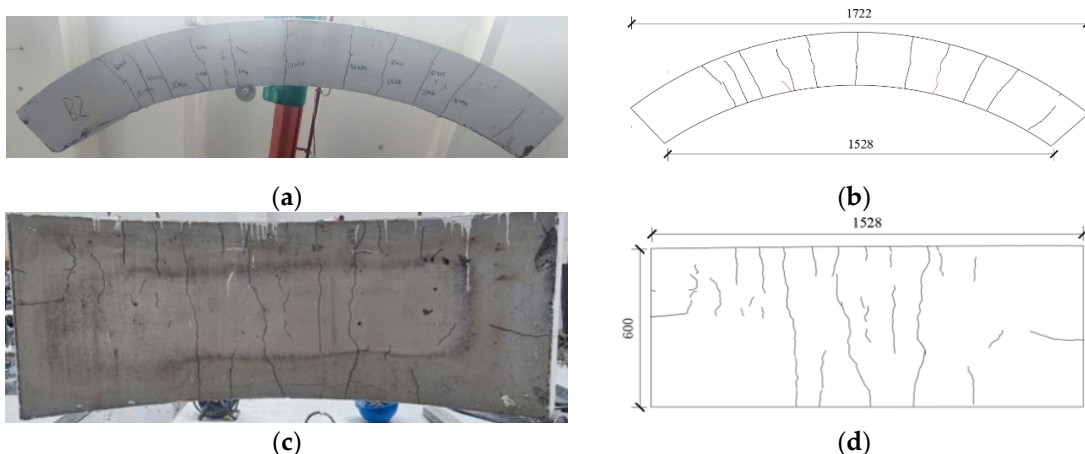

**Figure 8.** Failure modes of Segment RC—2: (**a**) Cracking pattern; (**b**) Crack pattern on the side surface; (**c**) Bottom surface; and (**d**) Cracking pattern on the bottom surface.

(3)    Segment HFRC−3

As shown in Figure 9, at 20 min, cracks appeared at the loading point, and the crack length exceeded 90 mm. At 30 min, cracks appeared in the mid-span, and water vapor began to appear at the support. At 50 min, a large number of cracks appeared on the pipe side, and the crack length at the loading point was 175 mm, and water stains appeared at the crack and cracks appeared at the support. At 70 min, water stains appeared on the vault, and the water vapor on the pipe side was concentrated in 20 min to produce cracks, the crack width at the loading point on the pipe side was 0.21 mm, and the crack in the span developed to the vault. At 180 min, the maximum crack width on the pipe side was 0.47 mm.

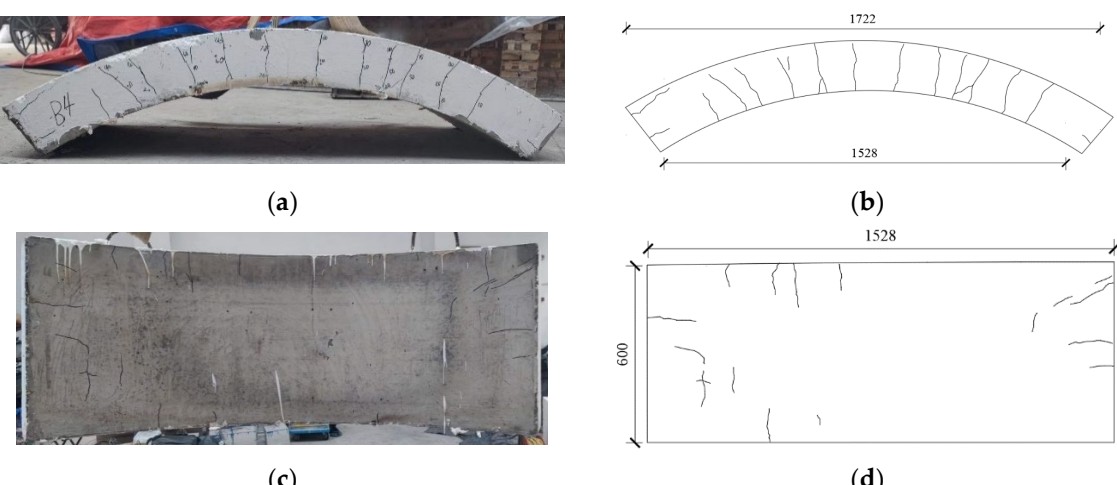

**Figure 9.** Failure modes of Segment HFRC—3: (**a**) Cracking pattern; (**b**) Crack pattern on the side surface; (**c**) Bottom surface; and (**d**) Cracking pattern on the bottom surface.

(4)    Segment HFRC−4

As shown in Figure 10, when the segment was loaded to 120 kN, there were no cracks on the sides, and, after the load was stabilized, the temperature began to rise. Heating for 20 min, cracks were first generated at the loading site, followed by the span. At 25 min, the crack mainly appeared between the loading point and the middle span, and the crack length exceeded 1/2 of the side of the segment. At 40 min, a large number of cracks began to appear on the side of the segment, and, at the same time, water stains and cracks appeared at the support. At 50 min, the blisters seeped out from the crack, and water vapor appeared

at the support. After 80 min, water stains began to appear in the vault, and penetrating cracks appeared at the loading point. After 120 min, a penetrating crack appeared in the middle of the span.

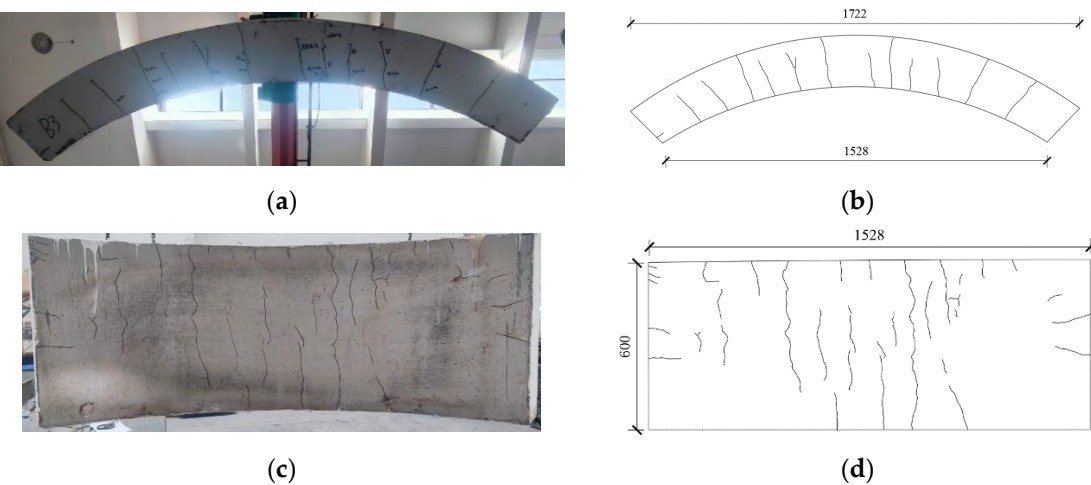

**Figure 10.** Failure modes of Segment HFRC—4: (**a**) Cracking pattern; (**b**) Crack pattern on the side surface; (**c**) Bottom surface; and (**d**) Cracking pattern on the bottom surface.

(5)　Segment HFRC−5

As shown in Figure 11, cracks appeared at the loading point when the segment was loaded to 150 kN, and the number of cracks on the tube side reached four at 180 kN. The temperature rose for 10 min, and cracks appeared in the span, the initial crack at the loading point extended, and the arch foot was crushed. At 30 min, a large number of cracks appeared on the pipe side, and the maximum crack length was 85 mm, and water stains appeared at the support. At 60 min, water vapor appeared at the crack on the side of the pipe, the length of the crack at the loading point exceeded 180 mm, and water stains appeared on the vault.

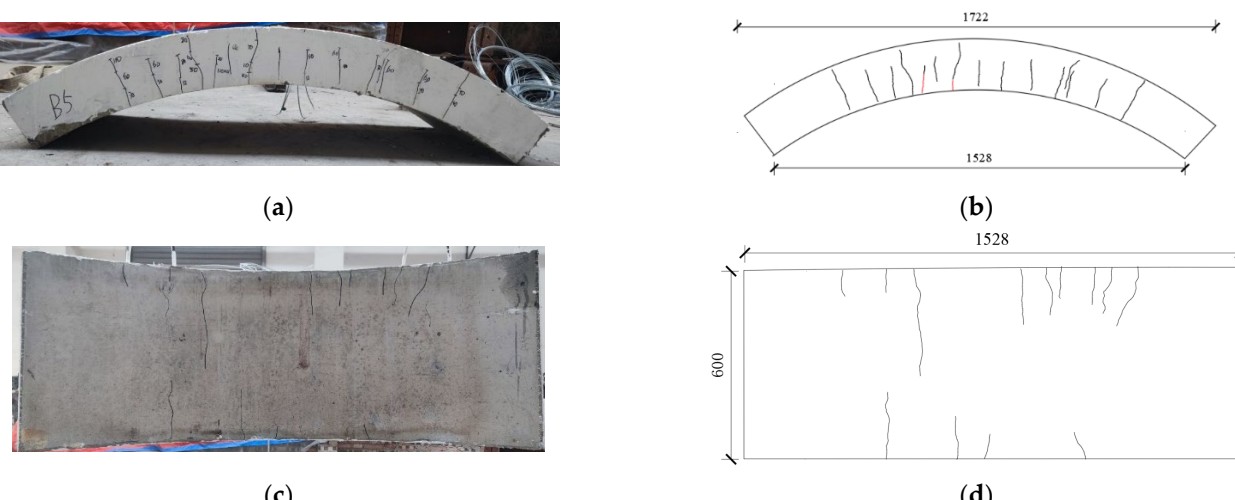

**Figure 11.** Failure modes of Segment HFRC−5: (**a**) Cracking pattern; (**b**) Crack pattern on the side surface; (**c**) Bottom surface; and (**d**) Cracking pattern on the bottom surface.

(6)　Comparative analysis

It can be seen from Figure 7 that, for the side of the segment, the crack was caused by the combination of vertical load and temperature stress, and was mainly distributed between the loading point and the span, which is similar to the stirrup spacing. In addition,

the loading point and the span middle crack developed rapidly. For vaults, cracks were mainly composed of pipe-side extension cracks, with a crack length of 10–30 mm. For the arch bottom, when the pre-loaded load was small, the arch bottom cracks were mainly distributed in the area without fire, and when the pre-loaded load was large, the through-length cracks appeared in the arch bottom loading points and spans. At the same time, under the action of a high temperature, a large number of dense cracks appeared.

For Figures 8–10, it can be seen that, with the increase in pre-loading, the number of cracks at a high temperature on the side of the segment increased significantly, and most of them were concentrated at the loading point. At the same time, due to the increase in the top compressive stress, the tensile stress generated by thermal expansion canceled each other out, so that the crack length decreased.

It can be seen from Figures 7 and 8 that, when the vertical load was small, the arch foot was swollen by fire and squeezed with the support, resulting in a large crack width or even crushing. Therefore, in the shield tunnel structure with a shallow buried depth, the reinforcement at the corner of the arch can be appropriately strengthened, so as to improve the strength at the foot of the arch.

Compared with the undoped segments, the cracks of the segments mixed with steel fibers and polypropylene fibers appeared later, the number of cracks increased, the average crack spacing decreased, the distribution of tensile stress was more uniform, and, at the same time, the crack bifurcation phenomenon was obvious. This is mainly due to the difference in the bearing capacity of the two cross-section types. Since the fibers are uniformly dispersed in the cross-section, the tensile forces subjected are distributed over the entire cross-section [19,20], showing a more uniform distribution of tensile stress and a smaller average crack spacing.

*3.2. Concrete Temperature*

The temperature–time curve of concrete in the whole process of segment heating up and down is shown in Figure 12. Due to the similar furnace temperature and small degree of bursting, the temperature of the arch surface in the segment did not undergo sudden temperature mutation. RC—1, RC—2, and HFRC—3 have maximum temperatures of 498 °C (112 °C), 550 °C (106 °C), and 501 °C (103 °C) on the bottom (10 mm) and top (160 mm) segments, respectively. At the same time, it can be seen from Figure 12 that there is a temperature platform at the concrete temperature measurement point of the vault, in which the farther away from the fire surface, the longer the temperature platform exists, and the greater the influence of water vapor evaporation on the segment.

According to GB 50016-2014 [21], the fire resistance limit of the load-bearing structure in the tunnel is as follows: when the temperature of the steel bar 25 mm from the bottom surface of the concrete exceeds 250 °C, or the temperature of the concrete surface exceeds 380 °C. As can be seen from Figure 12, when the fire duration time was about 110 min, the surface temperature of the concrete reached 380 °C, and there was no failure phenomenon on the segment with a small load level (60 kN), while there were more cracks on the side of the segment with a large load level (120 kN and 180 kN). It can be seen that GB 50016-2014 only takes temperature as the basis for determining the fire resistance limit of load-bearing structures in tunnels, which is more one-sided and conservative.

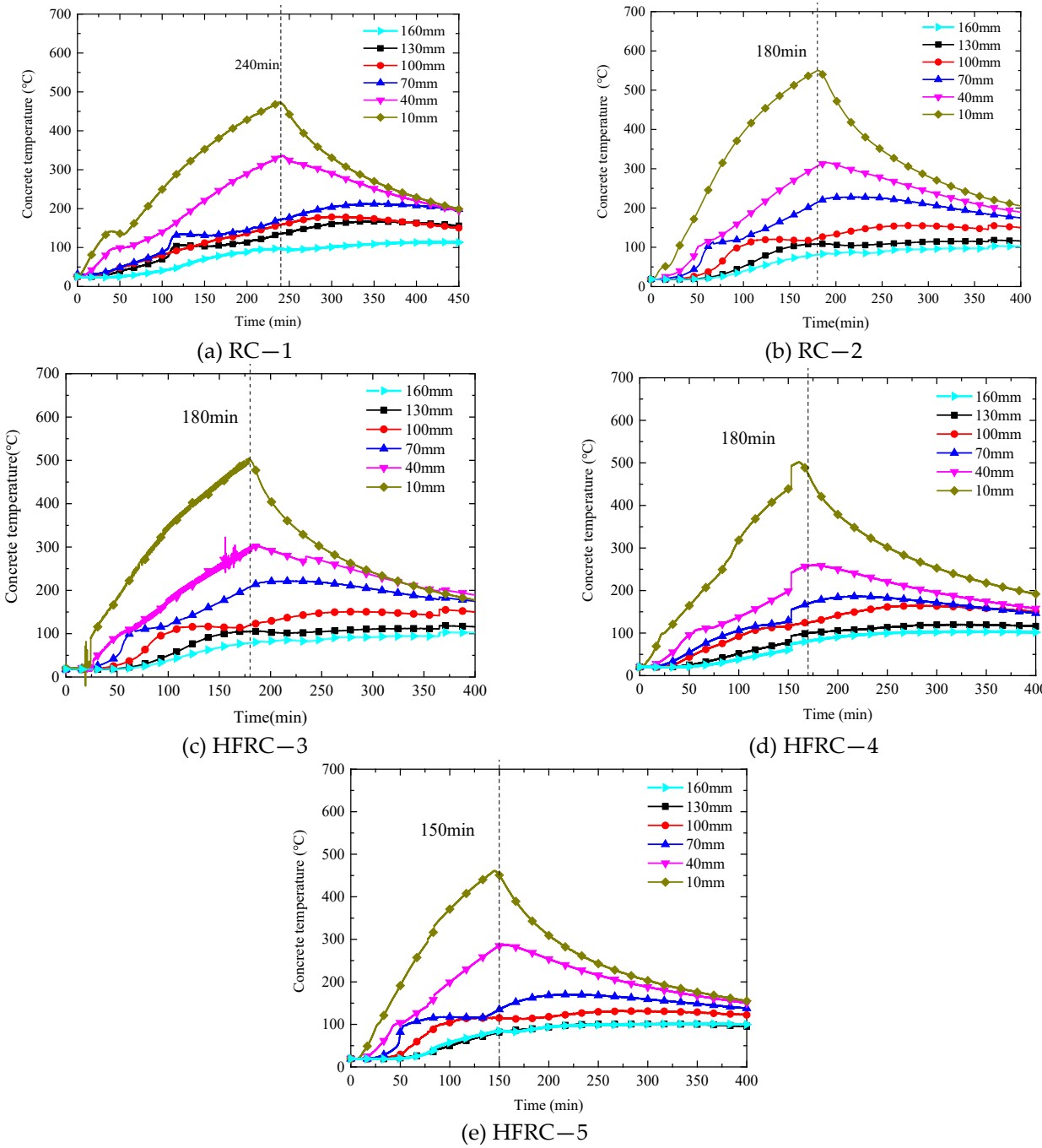

**Figure 12.** Time–temperature curves of segment sections: (**a**) RC—1; (**b**) RC—2; (**c**) HFRC—3; (**d**) HFRC—4; and (**e**) HFRC—5.

### 3.3. Steel Bar Temperature

Figure 13 shows the temperature–time curves of the steel bars of the five segments in the whole process of temperature ramping. It can be seen from the figure that the time corresponding to the highest temperature of the lower steel bar was close to the ceasefire time, and, due to the thermal inertness of the concrete, the maximum temperature of the upper steel bar had a certain delay.

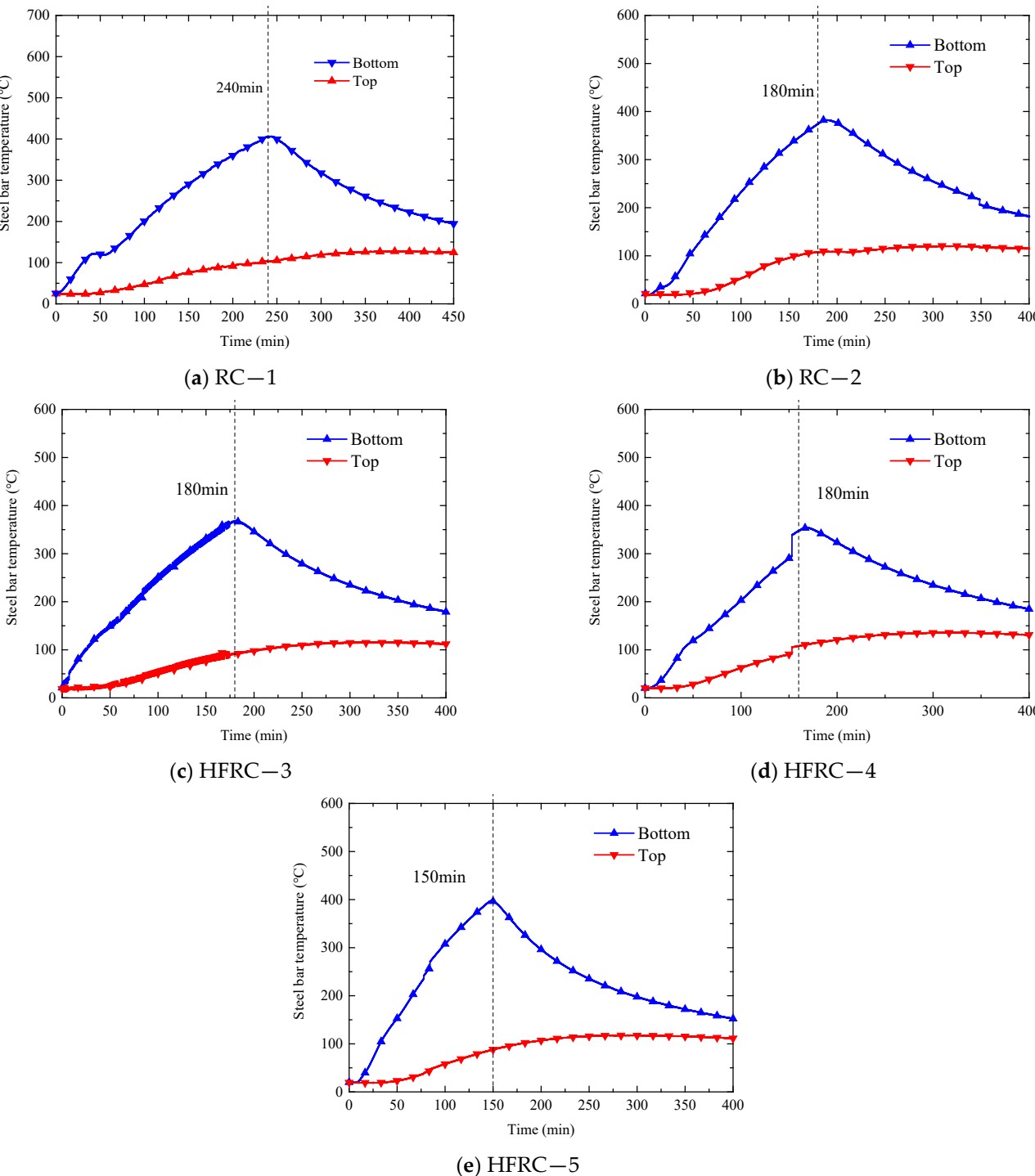

**Figure 13.** Time–temperature curves of segment steel bars: (**a**) RC—1; (**b**) RC—2; (**c**) HFRC—3; (**d**) HFRC—4; and (**e**) HFRC—5.

### 3.4. Deflection Behavior

Figure 14 shows the vertical displacement–time curves of the five concrete segments in the mid-span and loading point. where negative values represent downward deformation and positive values represent upward deformation.

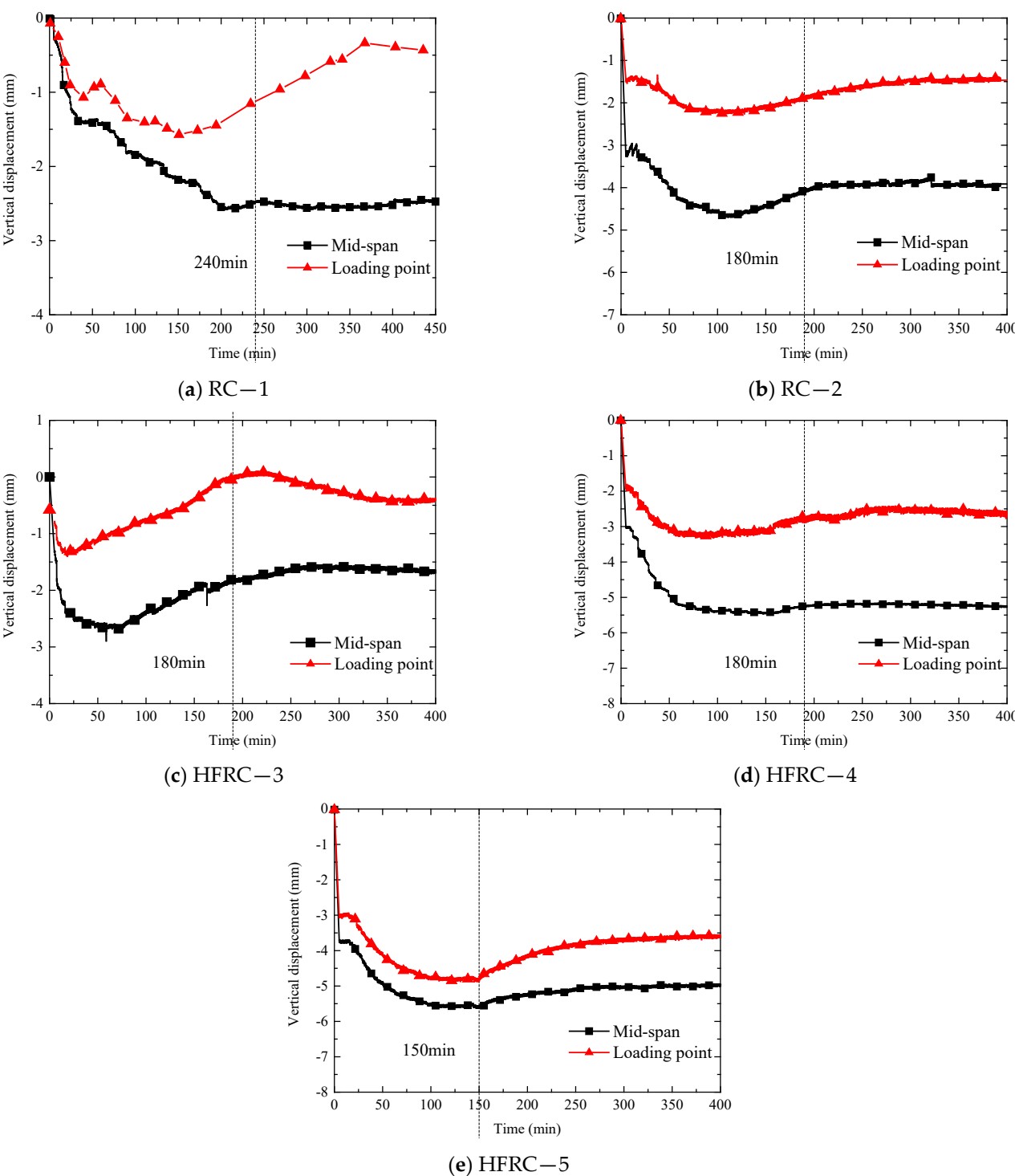

**Figure 14.** Vertical displacement–time curves: (**a**) RC—1; (**b**) RC—2; (**c**) HFRC—3; (**d**) HFRC—4; and (**e**) HFRC—5.

As shown in Figure 14a, in the first 30 min, the deformation of each point of segment RC—1 increased rapidly, and, after the temperature was stable, the trans-center displacement gradually slowed down and remained unchanged, while the loading point displacement recovered. At the end of the heating, the RC—1 segment's mid-span displacement and loading-point displacement were −2.56 mm and −1.12 mm, respectively. After the end of the cooling stage, the mid-span displacement of the segment was basically the same as the displacement after the end of the heating, indicating that the plastic deforma-

tion of the segment at a high temperature was large. As shown in Figure 14b,d, when the load level was large, the loading point displacement and the cross-center displacement showed the same development trend; that is, the initial heating increased rapidly, the temperature was stable, and the cooling stage remained basically unchanged.

Comparing Figure 14a with Figure 14b, it can be seen that, as the load level increased, the deformation also increased after the temperature rose. At the same time, the initial load level had a great influence on the initial displacement and crack development; therefore, for the segment with a deep buried depth, the bottom steel bar of the segment should be increased. Comparing Figure 14b with Figure 14d, it can be seen that, under the same load conditions, the fiber had less influence on the deformation of the segment.

### 3.5. Horizontal Support Reaction Force

The Horizontal support reaction force–time curves of the five segments are shown in Figure 15. As can be seen from the figures, the maximum horizontal force and development trend are related to the preloaded load values and fire duration. For segments RC-1 and HFRC−3, it can be seen that the horizontal support reaction force due to temperature stress was greater than the applied preload value, which was 90 kN and 82 kN, respectively. It is explained that, for the shield tunnel system with a shallow burial depth, the low stress level caused the transient thermal strain to have less influence on the thermal stress, resulting in a large horizontal force at the support. With segments of 120 kN and 180 kN, the maximum horizontal support reaction force was smaller to the applied load.

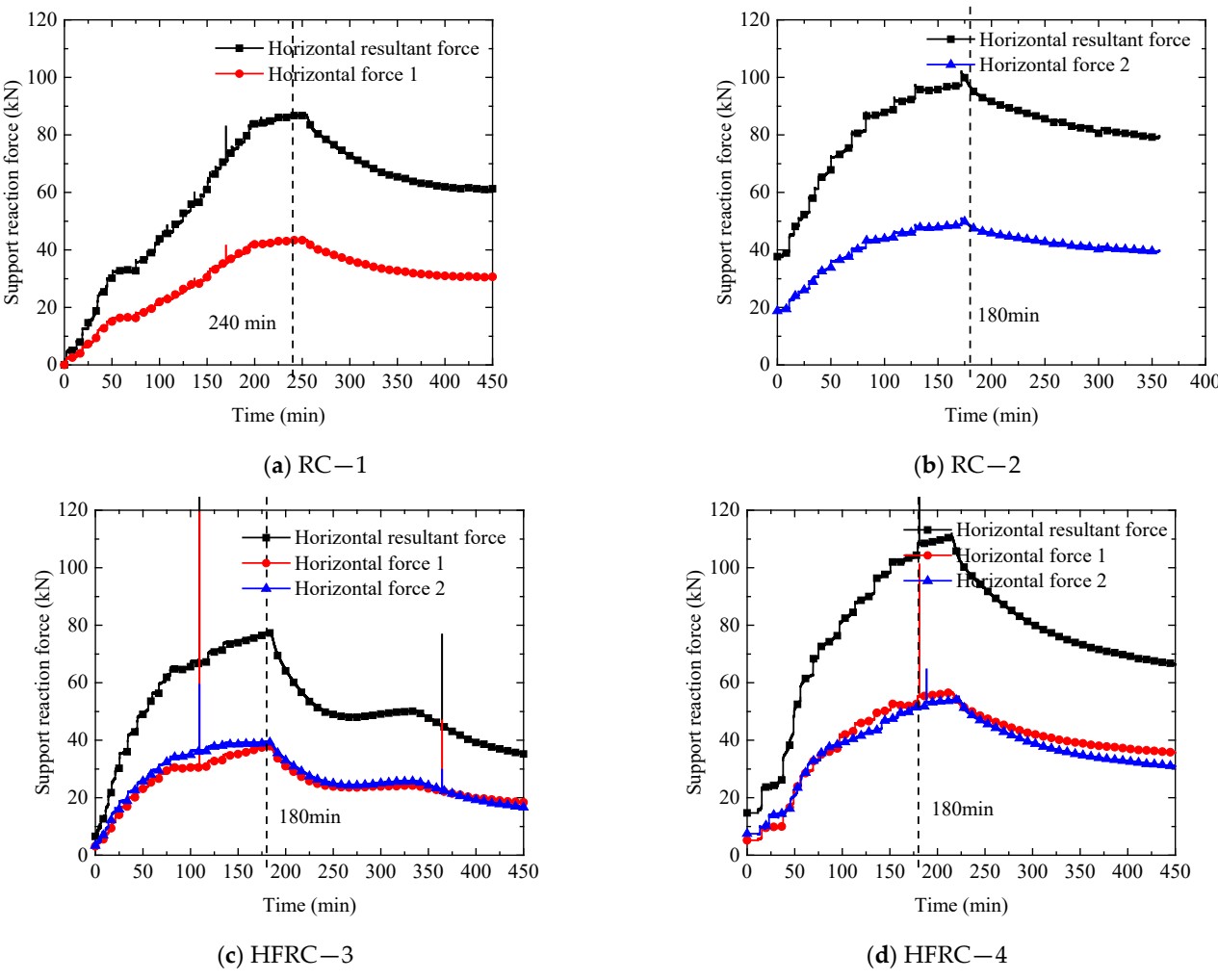

(**a**) RC−1

(**b**) RC−2

(**c**) HFRC−3

(**d**) HFRC−4

**Figure 15.** *Cont*.

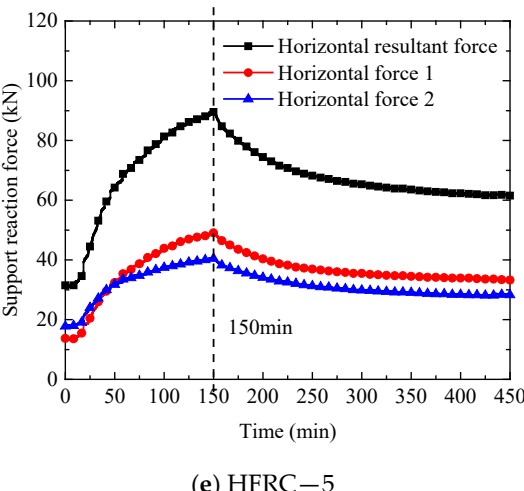

(**e**) HFRC—5

**Figure 15.** Horizontal support reaction force–time curves: (**a**) RC—1; (**b**) RC—2; (**c**) HFRC—3; (**d**) HFRC—4; and (**e**) HFRC—5.

## 4. Conclusions

This paper investigates the behavior of five concrete shield tunnel segments with two self-compacting concrete and three mixed-fiber (steel fiber and polypropylene fiber) self-compacting concrete segments subjected to fire. The main conclusions drawn are as follows:

(1) The type of fiber and pre-loading have an important influence on crack development in concrete segments. Compared with undoped segments, cracks in segments with steel fibers and polypropylene fibers appear later, and the average crack spacing decreases. With the increase in the pre-loaded load, the number of cracks at a high temperature on the side of the segment increases significantly, and most of them are concentrated at the loading point.

(2) Pre-loading has an important effect on the vertical deformation before and after the temperature rise. The initial load level has a great influence on the initial displacement and crack development. As the load level increases, so does the deformation after the temperature rise. Therefore, for segments with a deep buried depth, the steel bar at the bottom of the segment should be increased.

(3) The existing fire protection code only takes temperature as the basis for determining the fire resistance limit of load-bearing structures in the tunnel, which is more one-sided and conservative, and the influence of the initial load level should be considered.

(4) The horizontal support reaction force was affected by the preload and the fire condition, which shows the same trend as the furnace temperature. As the furnace temperature rises, the horizontal support reaction force increases.

**Author Contributions:** Conceptualization, Y.Z. and Y.W.; methodology, Z.R.; writing—original draft preparation, Y.Z.; writing—review and editing, Y.W.; funding acquisition, Y.Z. and Y.W. All authors have read and agreed to the published version of the manuscript.

**Funding:** This study was supported by the Joint Technology Transfer Center of Yancheng Polytechnic College, Yancheng Polytechnic College, the Excellent Young Backbone Teacher of Qinglan Project of Jiangsu Province, and the National Natural Science Foundation of China (Grant No. 51408594); "Double Carbon" Science and Technology innovation Special project—Social development science and technology demonstration project (KC22347); Innovation Ability Improvement Project of Science and Technology SMEs in Shandong Province (2022TSGC2170); Graduate Student Innovation Program of China University of Mining and Technology (KYCX22_2579). The authors gratefully acknowledge the support.

**Institutional Review Board Statement:** Not applicable.

**Informed Consent Statement:** Not applicable.

**Data Availability Statement:** Not applicable.

**Acknowledgments:** The authors would like to thank the funding support by Yancheng Polytechnic College.

**Conflicts of Interest:** The authors declare no conflict of interest.

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
