# Peer review of "Experimental Study on the Properties of Mixed-Fiber Concrete Shield Tunnel Segments Subjected to High Temperatures"

_fire, doi:10.3390/fire6010017_

Round 1

Reviewer 1 Report

The reviewer believes that this paper discusses a topic which is of high interest for research in the field of tunnel fire safety. The amount and quality of the presented work is satisfactory However, some improvements could be made as proposed in the following remarks:

- Table 2: Please explain the reasons why the load level of 180 kN was not tested at the RC case but only at the HFRC case. How will the comparison be conducted? In addition, an explanation is required on the determination of the fire durations. The round numbers (150, 180, 240 min) indicate that they were not determined by the time of failure of the specimens. So, how were they chosen and why? Finally, the combination of parameters (fire duration and load level) by varying both makes the comparisons difficult. For instance, the two RC segments were tested either for 3 hours with a high load of 120 kN or for 4 hours but with a lower load (60 kN). By varying both parameters simultaneously, determining the effect of each parameter is not straightforward. Please justify the variation of these parameters and the tested combinations.

- How was the load controlled during the test? Was it kept constant throughout the test or were load drops allowed due to crack development after heating?

- L.198-200: The thermal expansion of the specimens is restrained by the supports therefore the developed stresses should be compressive. The authors mention tensile stresses developed by the thermal expansion. Please explain the mechanism that causes tensile stresses or correct this part of the discussion.

- Section 3.2: As seen by the graphs in Figure 11, the adopted temperature-time curves are in all cases “lower” that the ISO 834 curve. However, tunnel fires usually result to fire curves more severe than the ISO 834 (e.g. RWS (Rijkswaterstaat) fire curve, or RABT-ZTV fire curve). Please explain the choice of the adopted exposure temperatures and justify the validity of the results considering the low severity of the adopted heat treatment.

- L.229-230: “it can be seen from Figure 12 … 380 oC” : Please rephrase this sentence because the meaning is not clear.

- Based on Figures 12 and 13 it can be seen that the maximum temperatures are similar regardless of the duration of the fire exposure. This is justified by the fact that the duration of 240 minute was with a fire curve of lower severity compared to others (see Figure 11) so despite the longer time the maximum temperatures were not higher than in the other cases. Hence, the parameter concerning the duration of the fire exposure was not properly investigated since it does not seem to play an effect in these tests. Please comment on this result in the manuscript.

- Section 3.6: Please explain how the horizontal force was measured.

- Apart from the heating rate which is given in detail by the temperature measurements, another significant parameter that affects the fire performance of concrete members (e.g. the spalling phenomenon that was not evidenced in this study) is the moisture content. Please provide some information about the moisture content of the specimens, e.g. any measurements if available or the initial water-to-cement ratio and the curing conditions.

Author Response

We thank the reviewers for their significant efforts and insightful comments, which have been extremely helpful in improving the quality of this paper, both in terms of writing style and technical issues.

The response to the reviewer's comment has been upload, Please see the attachment.

Reviewer 2 Report

The article is focused on evaluating the effect of fire on shield tunnel segments, which is an important topic that requires attention in current literature. The following comments require attention from the authors:

- In the abstract, the reviewer suggests the removal of the "etc" in line 13, since it does not provide an appropriate representation of what was studied in the work and leaves space open for interpretation.

- The term "fire time" can be replaced by "fire duration" for a more appropriate meaning. In Line 247 there is a dot before the word "where", the reviewer believes it should be a comma. The term “load size” should be replaced by “load level” in line 291. The paper is well written, even though a review could be done to fix some minor issues.

- The literature review and discussions can be improved. Most references are in Chinese and refer to local guidelines and to Master thesis. Several papers are published in relevant journals worldwide. The reviewer suggests that the authors improve the literature review and compare the results obtained to works published in literature. The reviewer has compiled some relevant and recent studies in literature, bellow. More references can be found at the references of the most recently published papers.

[1] Zhang Y, Ju JW, Zhu H, Yan Z (2020) A novel multi-scale model for predicting the thermal damage of hybrid fiber reinforced concrete. Int J Damage Mech 29:19–44. https://doi.org/10.1177/1056789519831554

[2] Agra RR, Serafini R, de Figueiredo AD (2021) Effect of high temperature on the mechanical properties of concrete reinforced with different fiber contents. Constr Build Mater 301:124242. https://doi.org/10.1016/j.conbuildmat.2021.124242

[3] Choumanidis, D., E. Badogiannis, P. Nomikos, and A. Sofianos. 2016. “The effect of different fibres on the flexural behaviour of concrete exposed to normal and elevated temperatures.” Constr. Build. Mater. 129 (Dec): 266–277. https://doi.org/10.1016/j.conbuildmat.2016.10.089

[4] Serafini, R., S. R. A. Dantas, R. P. Salvador, R. R. Agra, D. A. S. Rambo, A. F. Berto, and A. D. de Figueiredo. 2019. “Influence of fire on temperature gradient and physical-mechanical properties of macro-synthetic fiber reinforced concrete for tunnel linings.” Constr. Build. Mater. 214 (Jul): 254–268. https://doi.org/10.1016/j.conbuildmat.2019.04.133

[5] Rambo, D. A. S., A. de Blanco, A. D. Figueiredo, E. R. F. dos Santos, R. D. Toledo, and O. F. M. da Gomes. 2018. “Study of temperature effect on macro-synthetic fiber reinforced concretes by means of Barcelona tests: An approach focused on tunnels assessment.” Constr. Build. Mater. 158 (Jan): 443–453. https://doi.org/10.1016/j.conbuildmat.2017.10.046

- The methodology section of the paper should be improved. The fire curve adopted as reference must be presented in this section. Also, Section 3.2 must be presented in the methodology section, since it refers to the parameters that were employed and is not analyzed. The methodology should mention if cooling was controlled or not controlled.

- The fire curve adopted is not representative of fire events in underground tunnels. The most indicated fire curves are the hydrocarbon fire curve and the RWS fire curve given that tunnels are enclosed spaces and fire temperatures tend to increase faster. This must be clear in the article, especially given that the authors have chosen the ISO 934 fire curve as reference.

- The fire curves are different among the specimens tested, especially in terms of fire duration. The authors should include this information in the methodology section and state that the results were analyzed only up to ~150min to ensure similar heating patterns between the samples.

- The GB 50016-2014 considers the temperature on the surface and on the rebar as failure criteria. However, this is extremely conservative and does not consider the differences in terms of bearing capacity provided by FRC and RC sections. These prescriptive parameters do not aid in the calculation of the bearing capacity of sections by engineers and more appropriate guidelines are available for fire in TBM tunnels, such as the Italian guideline CNR DT 204.

- The behavior of FRC and RC sections under fire differs significantly. The major difference is because a considerable portion of the bearing capacity of sections is reduced as soon as the temperature of rebars increase over ~400 °C. This is also related to the concrete cover. However, this behavior is completely different for FRC sections since the reinforcement is dispersed in the concrete section. Therefore, it is important to remark the differences in terms of bearing capacity between the two section types and compare to the results obtained in this paper. Some papers indicated below can aid in enhancing this comparison:

[6] Serafini, R., de La Fuente, A., & de Figueiredo, A. D. (2021). Assessment of the post-fire residual bearing capacity of FRC and hybrid RC-FRC tunnel sections considering thermal spalling. Materials and Structures, 54(6), 1-18. https://doi.org/10.1617/s11527-021-01819-2 

[7] Di Carlo F, Meda A, Rinaldi Z (2018) Evaluation of the bearing capacity of fiber reinforced concrete sections under fire exposure. Mater Struct 51:154.

- The conclusions can be improved. Conclusion (3) is not a conclusion from the results obtained in the paper and should be removed. The conclusion regarding the horizontal support forces may be presented. 

Author Response

(The authors gave the same response as above.)
